# DF-ExpEnse: Diffusion Filtered Exploration for Sample Efficient Finetuning

Calvin Luo [1 2]    Chen Sun [2]    Shuran Song [1]

## Abstract

A natural recipe for intelligent robotic decision-making is initializing from pretrained generative control policies, which have summarized offline experience, and adapting them to self-collected online experience. We present DF-ExpEnse, an exploration technique that improves the quality of online experience collection, thus increasing finetuning sample-efficiency. DF-ExpEnse leverages the multimodal modeling capabilities of the generative control policy to create an expressive and tractably evaluatable candidate set. It then utilizes an ensemble of critics to identify the action that best balances quality with high exploration interest. In fleet settings, DF-ExpEnse further enables cross-agent communication to facilitate collaborative exploration as a group. DF-ExpEnse can be seamlessly integrated with existing strategies that finetune pretrained generative control policies via reinforcement learning. We experimentally validate consistent sample-efficiency benefits through DF-ExpEnse across a variety of manipulation and locomotion tasks, compared to default finetuning and alternative action selection schemes. Project can be found at `df-expense.github.io`.

## 1. Introduction

Crucial to increasing sample efficiency for robotic policy improvement is effective **exploration** during online experience. We identify two promising developments that naturally support the investigation and design of enhanced exploration strategies. First, advances in behavior cloning (Pomerleau, 1988; Bain & Sammut, 1995) through action-space generative modeling such as diffusion policies (Chi et al., 2023) have enabled the conversion of offline demonstrations into strong behavior priors. Such pretrained generative policies are attractive to apply reinforcement learning finetuning on

[1]Stanford University [2]Brown University. Correspondence to: Calvin Luo <calvinyl@stanford.edu>.

*Proceedings of the 43rd International Conference on Machine Learning*, Seoul, South Korea. PMLR 306, 2026. Copyright 2026 by the author(s).

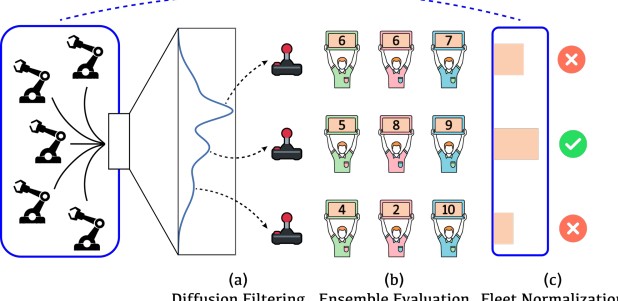

*Figure 1.* **DF-ExpEnse Overview.** At each timestep, DF-ExpEnse selects an exploratory action to execute by performing three steps. First, **(a)** filters the continuous action space by generating multiple samples from the diffusion policy. Then, **(b)** estimates exploration interest in each action with respect to quality and uncertainty using an ensemble. Lastly, **(c)** normalizes exploration interest across the fleet and selects the action with the maximum interest to execute.

top of (Ankile et al., 2025a; Ren et al., 2024; Wagenmaker et al., 2025), but are also capable of generating diverse modes of actions (Figure 1) suitable for exploration. Second, advances in *parallelization* either through simulation environments or large-scale real-world deployments (Agaskar et al., 2025; Schwall et al., 2020) naturally scale up online experience collection. Beyond simply increasing the quantity of data gathered, we believe such robotic fleets also hold potential in improving the quality of collected experience for efficient finetuning when used in a *collaborative* manner. In this work, we investigate how offline pretrained diffusion policies instantiated across robotic fleets can perform principled exploration as a collective group to improve decision-making finetuning in a sample-efficient manner.

We present **Diffusion Filtered Exploration via Ensembles (DF-ExpEnse)**, a technique that can be applied to improve exploratory action selection during online experience collection, and facilitate increased sample efficiency in finetuning a pretrained diffusion policy. In the design of DF-ExpEnse, we are guided by three motivating questions of interest:

1. *How can we **quantify** the exploration interest an agent has for an arbitrary action in a principled manner?*
2. *How can we tractably **identify** reasonable actions worth exploring amongst a continuous action space?*
3. *How can we achieve benefits from enabling parallelized deployments of agents to **collaborate** on exploration as a group rather than operating individually?*

Ideally, efficient exploration strategies should seek to balance high-quality actions along with uncertainty reduction. DF-ExpEnse utilizes an *ensemble of critics* as a natural mechanism for simultaneously **quantifying** both action quality as well as estimated uncertainty (Chen et al., 2017). Inspired by Upper Confidence Bounds (UCB) (Auer et al., 2002) action selection, DF-ExpEnse calculates exploration interest in an action as a linear combination of value estimation, computed as the minimum of the ensemble predictions, and critic disagreement, computed as the standard deviation of the ensemble predictions. DF-ExpEnse can reuse existing critic ensembles from the underlying reinforcement learning framework. In robotic decision-making, which features a continuous action space, finetuning is often performed through policy gradient techniques, which commonly utilize an ensemble of critics to evaluate experience and guide policy updates. However, beyond limiting their utilization to the optimization step alone, DF-ExpEnse proposes extending their usage to online inference as well to quantifiably estimate exploration interest of an action before selection.

Whereas using ensembles to evaluate and rank every possible action before choosing the optimal one for execution is tractable in environments with discrete action spaces (Chen et al., 2017; Jiang et al., 2023), the continuous action space commonly featured in robotic decision-making makes this exhaustive search infeasible. How can we tractably **identify** the action that best balances quality against uncertainty from a continuous action space? DF-ExpEnse leverages the *multimodal modeling capability* of the diffusion policy during online inference to propose a set of action candidates which 1) exhibit broad coverage of the action space when helpful, 2) are of a base admissible quality to consider executing, and 3) are of a tractable size to enumeratively evaluate. DF-ExpEnse thus essentially treats the modes estimated by the diffusion policy, approximated by sampled outputs, as a reasonable constrained space over which to search for an exploratory action worth attempting in the environment.

Rapid simulator advancements and increasing amounts of real-world robotic deployments naturally enable parallelized online experience collection. Commonly, each member of a robotic fleet performs inference independently; however, this naive approach may result in agents simultaneously gathering highly similar or redundant data. When instantiated in a fleet setting, DF-ExpEnse performs real-time *cross-fleet communication* to encourage the group as a whole to **collaborate** on collecting high-quality and diverse online experience. In particular, DF-ExpEnse exposes the critic ensemble predictions of each agent across the fleet at each timestep. Then, before action selection, each individual agent normalizes their exploration interest estimates using statistics from the entire fleet. This helps to rescale locally considered candidate actions according to global fleet information, allowing the identification and selection of actions that are relatively more uncertain or valuable contextualized by the situations and considerations of the rest of the group.

In a return to the central motivating questions, DF-ExpEnse uses a critic ensemble to quantify exploration interest, a sample-based approximation of the diffusion policy's multimodal distribution for tractable action evaluation, and real-time communication across the fleet to effectively contextualize candidate actions and facilitate group-collaborative exploration. DF-ExpEnse is a general exploration technique, and can be directly applied on top of a variety of methods that finetune diffusion policies through reinforcement learning to improve online experience collection quality. In our experiments, we combine DF-ExpEnse with DSRL (Wagenmaker et al., 2025) as well as ResFiT (Ankile et al., 2025a), and perform evaluations on manipulation tasks from the RoboMimic suite (Mandlekar et al., 2021), locomotion tasks from OpenAI Gym (Brockman et al., 2016), and bimanual manipulation tasks from DexMimicGen (Jiang et al., 2025). We consistently find that DF-ExpEnse achieves higher sample efficiency against vanilla reinforcement learning finetuning techniques as well as alternative exploration baselines. We also perform ablations on all three key components of our method: the size of the ensemble, the size of the sampled action set, and the size of the fleet. We empirically verify DF-ExpEnse to be a robust technique that can meaningfully improve online exploration behavior and the sample efficient finetuning of pretrained generative control policies.

## 2. Background and Related Work

**Finetuning Diffusion Policies with Reinforcement Learning:** Diffusion policies (Chi et al., 2023) have enabled the learning of powerful robotic decision-making models from behavior cloning on offline data. However, we also wish to enable a decision-making agent to continuously improve itself with respect to self-collected *online* experience (Silver & Sutton, 2025). As such, utilizing reinforcement learning to finetune pretrained diffusion policies is of interest. ResiP (Ankile et al., 2025b) and ResFiT (Ankile et al., 2025a) use a residual policy (Silver et al., 2018) that learns to adjust the output of a frozen diffusion policy via reinforcement learning. DPPO (Ren et al., 2024) treats the denoising procedure of action sampling as its own inner Markov Decision Process, and finetunes the weights of the diffusion policy directly with respect to environment feedback. DSRL (Wagenmaker et al., 2025), like the residual approach, keeps the pretrained diffusion policy frozen but learns an input noise selection policy through reinforcement. Our work seeks to increase sample efficiency in combining reinforcement learning updates with a pretrained diffusion policy by improving exploration during online experience collection, and can be seamlessly integrated on top of these techniques.

**Ensembles for Uncertainty Estimation:** Upper-confidence bound algorithms (Auer et al., 2002) that utilize ensembles

to estimate uncertainty have been used in prior works for improved exploration (Chen et al., 2017; Lee et al., 2021) as well as generalization (Jiang et al., 2023; Shi et al., 2024). We apply a similar UCB action selection strategy to identify and execute action candidates that best balance quality with uncertainty. However, because the ensembles are used to evaluate individual actions at a time, such works are usually applied over environments with discrete action spaces. Prior works (Lee et al., 2021; Shi et al., 2024) bypass this restriction in continuous action spaces by utilizing an ensemble of policies in tandem to generate a discrete set of proposal actions. In this work, we show that the multimodal modeling property of diffusion policies can filter the continuous action space into a set of reasonable candidates to consider, with broad coverage over the action space when necessary.

**Inference Time Selection:** Sampling a set of action candidates from the policy before selecting one for execution allows for more intentional, targeted behavior. V-GPS (Nakamoto et al., 2024) generates multiple prospects from a pre-trained policy and samples one to execute weighted by their Q-value estimate. However, the goal of V-GPS is to steer a generalist robotic policy for downstream deployment, whereas our work seeks to achieve *explorative* experience collection and sample-efficient finetuning of the policy. Rather than sampling, EMaQ (Ghasemipour et al., 2021) and Q-Chunking (Li et al., 2026) select the action candidate with the maximum estimated Q-value. Selecting the maximum value can ignore exploration interest, and can lead to a less robust and generalizable policy over finetuning iterations. We compare our exploration technique against a Max-Q selection baseline and demonstrate improvements with respect to downstream finetuning sample-efficiency.

**Fleet Learning:** As parallelization capabilities improve in simulators (Makoviychuk et al., 2021) and real-world robotic deployments increase (Kalashnikov et al., 2021; Herzog et al., 2023), how to effectively harness such robotic fleets for enhanced policy learning is of great interest. Prior works have leveraged massive parallelization to quickly learn locomotion policies that transfer to the real world (Rudin et al., 2022). FLEET-MERGE (Wang et al., 2024) describes how merging policies across a fleet can still enable fleet-level learning without explicitly storing data in a centralized manner. SAPG (Singla et al., 2024) splits large-scale parallelized environments into chunks with separate policies, and aggregates their data through importance sampling to update a leader policy. In this work, we investigate how collaborative exploration as a group can be achieved for sample-efficient finetuning, by enabling real-time communication across the fleet when performing action selection.

## 3. Method

We present Diffusion Filtered Exploration via Ensembles (DF-ExpEnse), which can be applied on top of reinforce-

ment learning techniques that finetune offline pretrained generative control policies to improve their exploration and sample efficiency. We then describe how DF-ExpEnse can be extended when utilized in a fleet setting to communicate across agents and facilitate exploration as a collective group.

### 3.1. Diffusion Filtered Exploration via Ensembles (DF-ExpEnse)

At each timestep of online experience collection, DF-ExpEnse utilizes two model components: the diffusion policy $\pi_\theta^{dp}(\cdot)$ initially instantiated with offline behavior cloning pretraining, and a $K$-sized ensemble of Q-value functions $Q_{[1...K]}(\cdot)$, which can be randomly initialized. The critic ensemble, which is normally utilized during policy gradient updates, can be reused during this online inference step to help identify actions of exploratory interest to the agent.

At each timestep, we generate a set of $M$ candidate outputs conditioned on the current state $s$ using i.i.d. sampling:

$$[\boldsymbol{a}_1, \ldots, \boldsymbol{a}_M] \sim \pi_\theta^{dp}(\boldsymbol{a} \mid \boldsymbol{s}). \tag{1}$$

We then compute exploration interest of each candidate action in the set as a linear combination of their value estimate and the critic disagreement, as evaluated by the ensemble:

$$e_m = \min\left(Q_{[1...K]}(\boldsymbol{a}_m, \boldsymbol{s})\right) + \alpha * \text{std}\left(Q_{[1...K]}(\boldsymbol{a}_m, \boldsymbol{s})\right) \tag{2}$$

where the action's estimated value is represented by the minimum of the critic ensemble $\min\left(Q_{[1...K]}(\boldsymbol{a}_m, \boldsymbol{s})\right)$ to reduce potential overestimation, and its uncertainty is estimated via critic disagreement as the standard deviation over the ensemble's predictions. This is essentially a min-value version of UCB (Auer et al., 2002). Intuitively, if an action is roundly agreed upon to be of good quality by all critics in that the minimum estimation is high, but there is substantial disagreement amongst the critics as to what the exact value is, it is most likely a high-quality but underexplored action worth executing (Chen et al., 2017). Here, $\alpha$ is a hyperparameter that controls how much the agent considers uncertainty when computing an action's exploration interest.

Lastly, DF-ExpEnse selects the action candidate with the maximum estimated exploration interest for execution:

$$\boldsymbol{a}^\star \leftarrow \underset{\boldsymbol{a} \in [\boldsymbol{a}_1, \ldots, \boldsymbol{a}_M]}{\arg\max} [e_1, \ldots, e_M]. \tag{3}$$

The online experience collected by DF-ExpEnse can be directly utilized by off-the-shelf reinforcement learning techniques to finetune the agent. We note that when $\alpha = 0$, the agent performs exploitative behavior, consistently selecting the action candidate with the highest estimated value at each timestep. This is equivalent to what is described in EMaQ (Ghasemipour et al., 2021) and Q-Chunking (Li et al., 2026); we treat this greedy strategy as a natural baseline, and provide extensive experimental comparisons against it.

## 3.2. Behavior Cloning Sampling Regularization (BC-SR)

At heart, DF-ExpEnse relies on the multimodal modeling capability of the diffusion policy to constrain the search space of considered actions to a tractable size, helping the critic ensemble to focus its evaluation on candidates with a base admissible quality, and a broad coverage of the action space. However, over multiple rounds of finetuning, the agent may naturally begin to converge on fewer action modes. This can negatively affect exploration; in the extreme case where all candidates were sampled from the same mode, performing more principled action selection becomes futile. Thus, in order to foster continued exploration throughout finetuning we propose regularizing the set of action candidates with samples from the initial offline-pretrained diffusion policy, denoted by $\pi_{\theta_{\text{init}}}^{dp}(\cdot)$, which we expect to preserve more multimodal action modeling. Concretely, for a regularization integer $p \leq M$, we override the action candidate set with:

$$[\hat{\boldsymbol{a}}_1, \ldots, \hat{\boldsymbol{a}}_p] \sim \pi_{\theta_{\text{init}}}^{dp}(\boldsymbol{a} \mid \boldsymbol{s}), \tag{4}$$

resulting in a final considered action candidate set of:

$$[\hat{\boldsymbol{a}}_1, \ldots, \hat{\boldsymbol{a}}_p, \ldots, \boldsymbol{a}_M]. \tag{5}$$

We term this regularization technique **Behavior Cloning Sampling Regularization (BC-SR)**. Intuitively, BC-SR can be thought of as an inference-time technique similar in spirit to BC regularization (Fujimoto & Gu, 2021; Zhao et al., 2022; Nair et al., 2020; Goecks et al., 2019). Whereas BC regularization in reinforcement learning is a loss term that encourages the weights of the policy to remember how to generate samples from the offline dataset during the optimization procedure, here we regularize the set of candidate actions to consider for exploration directly, using generated samples from a BC model trained only on the offline dataset. Thus, BC-SR encourages the agent not to forget multimodal priors from offline pretraining during online inference. This approach is similar to the Actor Proposal step of IBRL (Hu et al., 2024) with two differences: firstly, the policy self-produces proposals through its own previous checkpoint rather than utilizing two separate models, effectively performing bootstrapped self-improvement, and secondly, the policy is instantiated as a diffusion policy with expressive multimodal modeling capabilities, enabling the sampling of a potentially more diverse action candidate set to consider.

## 3.3. DF-ExpEnse with Fleet Normalization

Thus far, DF-ExpEnse details how an individual agent can improve exploration quality by leveraging the multimodal nature of the generative policy combined with a critic ensemble during online inference. However, in the era of robotic fleets and highly parallelizable simulators, collecting online data for finetuning can be done in an increasingly

synchronous manner. In particular, such developments allow DF-ExpEnse to access multiple agents deployed in parallel.

Robotic fleets naturally increase the quantity of online data collection; in this work, we investigate strategies to also improve the quality of such collected experience. Whereas robotic fleets commonly deploy their agents independently from each other, we believe that there is potential in achieving more effective exploration as a group by enabling cross-fleet communication. By enabling communication between agents in a real-time, dynamic manner, they can collaborate to avoid exploring or experiencing redundant trajectories.

We thus describe the behavior of DF-ExpEnse in a fleet setting, to further improve collective exploration as a group. The key insight is that having each agent share their individual critic ensemble prediction terms with other members in real-time as summarized statistics can help contextualize the exploration interest of each candidate set against the rest of the fleet, and lead to improved collective decision-making.

Concretely, for a parallelized fleet of size $N$, DF-ExpEnse first generates a set of $M$ candidate outputs per individual agent, indexed by $n \in [1, \ldots, N]$, conditioned on their respective currently observed state $\boldsymbol{s}_n$, using i.i.d. sampling:

$$[\boldsymbol{a}_{n,1}, \ldots, \boldsymbol{a}_{n,M}] \sim \pi_\theta^{dp}(\boldsymbol{a} \mid \boldsymbol{s}_n). \tag{6}$$

At each timestep, we thus consider a set of $N \times M$ actions, aggregated across all agents in the fleet. DF-ExpEnse has each agent then calculate their individual value estimates and uncertainty scores using the critic ensemble as before:

$$v_{n,m} = \min\left(Q_{[1\ldots K]}(\boldsymbol{a}_{n,m}, \boldsymbol{s}_n)\right) \tag{7}$$

$$d_{n,m} = \text{std}\left(Q_{[1\ldots K]}(\boldsymbol{a}_{n,m}, \boldsymbol{s}_n)\right). \tag{8}$$

However, in the fleet setting, we further normalize each exploration interest component with respect to their respective terms aggregated from all candidate actions across the fleet:

$$\bar{v}_{n,m} = \frac{v_{n,m} - \text{avg}([v_{1,1}, \ldots, v_{N,M}])}{\text{std}([v_{1,1}, \ldots, v_{N,M}])} \tag{9}$$

$$\bar{d}_{n,m} = \frac{d_{n,m} - \text{avg}([d_{1,1}, \ldots, d_{N,M}])}{\text{std}([d_{1,1}, \ldots, d_{N,M}])} \tag{10}$$

$$\bar{e}_{n,m} = \bar{v}_{n,m} + \alpha * \bar{d}_{n,m} \tag{11}$$

where exploration interest $\bar{e}_{n,m}$ is now the linear combination of fleet-normalized value and critic disagreement predictions. Then, for each fleet agent indexed by $n \in [1, \ldots, N]$, the action candidate with the maximum estimated exploration interest after fleet normalization is selected and run:

$$\boldsymbol{a}_n^\star \leftarrow \underset{\boldsymbol{a} \in [\boldsymbol{a}_{n,1}, \ldots, \boldsymbol{a}_{n,M}]}{\arg\max} [\bar{e}_{n,1}, \ldots, \bar{e}_{n,M}]. \tag{12}$$

In Section 4.7 we empirically find that using fleet normalization provides sample-efficiency benefits beyond running a fleet of individual DF-ExpEnse agents in mutual isolation.

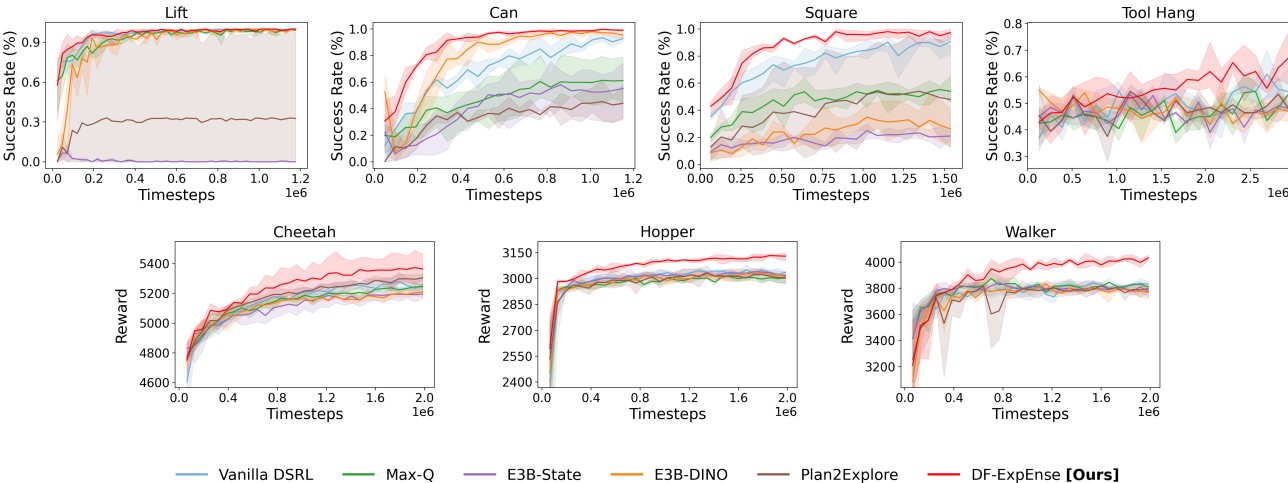

*Figure 2.* **Baseline Comparisons**. We compare DF-ExpEnse against vanilla DSRL, a Max-Q selection scheme, as well as other exploration baselines on RoboMimic manipulation and OpenAI Gym locomotion tasks, averaged over three random seeds. DF-ExpEnse consistently achieves superior sample-efficiency compared to other baselines, achieving a higher average success rate or reward score given the same amount of online experience timesteps. The benefits over vanilla DSRL, which DF-ExpEnse extends, demonstrates that using DF-ExpEnse can meaningfully increase principled exploration ability. Furthermore, as Max-Q also selects an action from a sampled candidate set, we demonstrate that the additional design decisions behind DF-ExpEnse directly correspond to performance improvements. DF-ExpEnse also outperforms techniques that utilize additional components, such as dynamics models, without needing additional components itself.

## 4. Experiments

We study how the exploration enhancements provided by DF-ExpEnse can aid sample-efficiency in finetuning a pretrained diffusion policy through reinforcement learning. As DF-ExpEnse is a technique for collecting experience during online inference, it can be seamlessly integrated with a variety of reinforcement learning finetuning techniques that perform the underlying policy optimization. Furthermore, DF-ExpEnse can be enabled or disabled at will; during inference rollouts for evaluation, when we seek to assess the quality of the finetuned policy directly, DF-ExpEnse is not used to ensure fair comparison with alternate techniques. No additional compute is utilized at test-time when evaluating policies finetuned via DF-ExpEnse against other baselines.

We begin by describing the choice of reinforcement learning algorithms used in our experiments, the tasks and environments chosen, the settings of the model components used, and the standard hyperparameters that DF-ExpEnse utilizes.

### 4.1. Reinforcement Learning Algorithms

We select DSRL (Noise Aliasing) (Wagenmaker et al., 2025) as the main underlying technique to perform reinforcement learning finetuning of a pretrained diffusion policy. DSRL freezes the pretrained diffusion policy and only interfaces with it via deterministic sampling, such as DDIM (Song et al., 2021). It performs reinforcement learning finetuning of the overall agent by optimizing a noise selector, implemented as a Soft-Actor Critic (SAC) agent (Haarnoja et al., 2018), that generates input noise to pass into the diffusion

model's deterministic sampling procedure. The code implementation of SAC used by DSRL, and thus reused in our experiments, is from Stable-Baselines3 (Raffin et al., 2021).

DF-ExpEnse extends vanilla DSRL only during the online experience collection step. In standard DSRL, the agent simply samples a singular action and directly executes it; with DF-ExpEnse, the executed action is selected from a sampled batch of action candidates according to exploration interest. We maintain the same quantity of data collection and keep the same underlying DSRL optimization scheme with respect to collected data to ensure fair comparison. As such, benefits in performance over vanilla DSRL can directly be attributed to an increase in quality of the data being collected, showcasing the benefits of careful action selection with respect to exploration. We provide a detailed tabulation of DSRL hyperparameters in Appendix A, where any deviations are denoted with an asterisk. A pseudocode of DF-ExpEnse with DSRL is provided in Appendix G.

We also investigate the integration of DF-ExpEnse with ResFiT (Ankile et al., 2025a), an alternate reinforcement learning finetuning technique which instead optimizes a residual that adjusts a base action prediction. As with DSRL, DF-ExpEnse only extends ResFiT during online experience collection; all updates are performed with the same hyperparameters and settings as vanilla ResFiT. A pseudocode of DF-ExpEnse with ResFiT is provided in Appendix H. We therefore demonstrate that DF-ExpEnse is a general online exploration technique that can be seamlessly combined with a variety of different strategies for updating pretrained generative control policies via reinforcement learning finetuning.

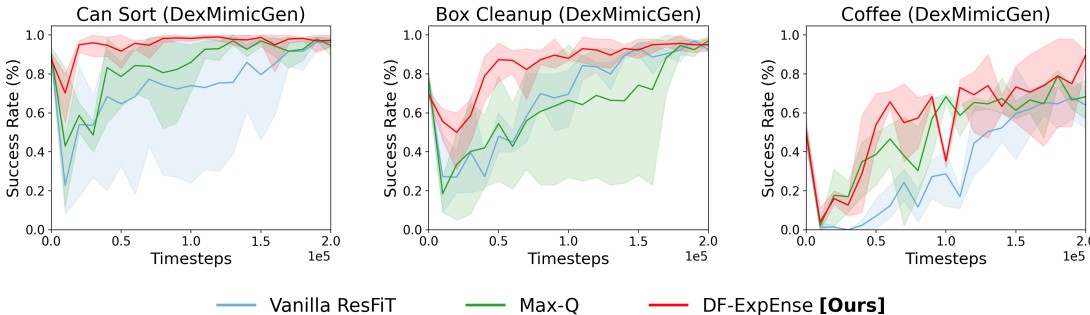

*Figure 3.* **ResFiT Baseline Comparisons**. We integrate DF-ExpEnse with ResFiT, and compare against a Max-Q selection scheme and vanilla ResFiT on DexMimicGen bimanual manipulation tasks, averaged over three random seeds. DF-ExpEnse continues to demonstrate sample-efficiency benefits compared to other baselines, achieving higher average success rates given the same amount of online experience. The successful application to an alternate reinforcement learning finetuning strategy highlights the general nature of DF-ExpEnse.

## 4.2. Tasks and Environments

**RoboMimic Environment (Manipulation):** We report results on the Lift, Can, Square, and Tool Hang tasks from the RoboMimic suite. These tasks are listed in order of difficulty, as described by the RoboMimic authors (Mandlekar et al., 2021). In particular, Tool Hang is the most complex, long-horizon task, where the robot must first assemble a frame through targeted insertion before picking up a tool and hanging it on the constructed frame. By default, RoboMimic provides sparse 0-1 rewards based on the success of the entire trajectory. For experiments on these tasks, we report success rate against environment timesteps collected throughout the duration of policy finetuning.

**OpenAI Gym Environment (Locomotion):** We report results on the HalfCheetah-v2, Hopper-v2, and Walker 2D-v2 locomotion tasks. Evaluation is reported as a numerical score computed by a ground-truth reward function against environment timesteps collected throughout finetuning.

**DexMimicGen (Bimanual Manipulation):** We report results on the Can Sort, Box Cleanup, and Coffee bimanual manipulation tasks from DexMimicGen, for ResFiT-based experiments. Evaluation is computed as binary success rates of overall trajectories logged against collected timesteps.

In evaluations for all tasks across all environments, numerical results are reported as averages over 100 rollouts, generated purely from the agent throughout stages of finetuning. DF-ExpEnse is disabled during test-time inference; it is only utilized when collecting online experience for optimization.

## 4.3. Base Diffusion Policy Implementations

In applying DF-ExpEnse as an exploration technique on top of DSRL, we reuse the same open-sourced pretrained diffusion policy checkpoints as in DSRL whenever available, for consistency. As noted in the DSRL manuscript, these checkpoints are themselves reused from the DPPO (Ren et al., 2024) public release, with the exception of the RoboMimic

Square checkpoint which was separately pretrained to increase the number of denoising steps. In our experiments, the only base policy we do not reuse from DSRL is the Tool Hang checkpoint, as Tool Hang was not a task featured in the original DSRL manuscript or codebase. We thus train our own base policy, which we implement as a flow policy (Black et al., 2024) with an output action horizon of size 8, where sampling is performed through 10 Euler integration steps. We note that DSRL by default also works with flow policies due to their inherent deterministic sampling procedure, and can be optimized in the same manner.

For experiments that integrate DF-ExpEnse with ResFiT, we reuse the same pretraining strategy as in the ResFiT manuscript to procure initial diffusion policy checkpoints. Such pretrained policies condition not only on proprioception but also vision inputs from all available RGB camera views. The base policy is trained to utilize 100 denoising steps, and generates sampled action sequences of length 16.

## 4.4. Default DF-ExpEnse Settings

DF-ExpEnse relies on three components: the current policy to sample a set of action candidates, a critic ensemble to evaluate exploration interest, and the fleet to communicate and receive statistics from its peers for more group-aware, collaborative action selection. All of these components already naturally occur in the DSRL framework; DF-ExpEnse simply exploits them during online inference for more principled selection of an exploratory action. By default, DSRL features a critic ensemble that is used to optimize the SAC noise selector agent. DSRL also utilizes a parallelized fleet to scale parallelized experience collection, but the vanilla setting deploys each agent independently from each other.

The main hyperparameters that adjust DF-ExpEnse's behavior thus pertain to these three components. Across all tasks and experiments, unless otherwise specified, we use a candidate action sample size of 3, a critic ensemble of size 10, and a fleet size of 4. We note that the DSRL by default uses a

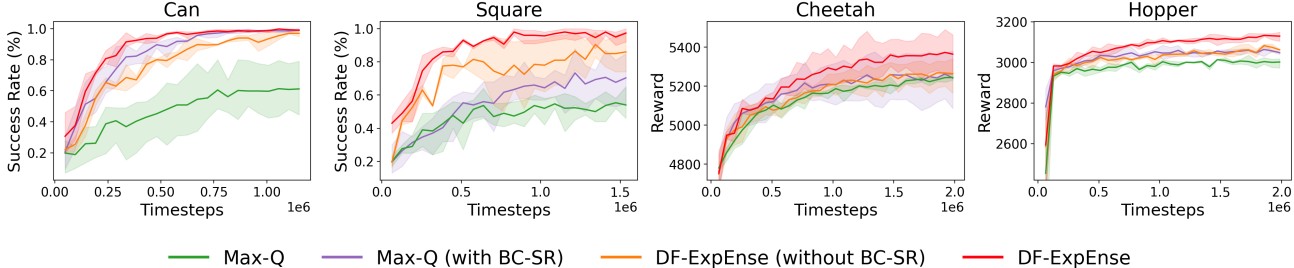

*Figure 4.* **On the Importance of BC-SR**. We demonstrate that BC-SR is a generally applicable regularization technique for augmenting the sampled action candidate set to encourage continued multimodal sampling. Removing BC-SR from DF-ExpEnse consistently decreases sample efficiency, while adding BC-SR to Max-Q significantly improves performance. All results are averages over three seeds.

fleet size of 4 for experience collection and a critic ensemble of size 2 for optimization. Whereas that may suffice for optimization purposes; we found that when we reuse the critic ensemble during online inference through DF-ExpEnse, a larger ensemble size induced greater performance benefits.

Furthermore, DF-ExpEnse utilizes BC-SR with a regularization amount of $p = 1$ for all experiments. Thus, at each timestep, the agent also considers a singular action sampled from the initial pretrained diffusion policy. In practice, because DSRL does not update the underlying pretrained diffusion policy's weights, this is generated by simply bypassing the learned noise selector network and directly providing a sampled Gaussian noise vector as input. Lastly, we use a critic disagreement parameter of $\alpha = 0.5$ in all experiments.

### 4.5. Baseline Comparisons

We report evaluations throughout the finetuning procedure across RoboMimic and OpenAI Gym tasks in Figure 2, where curves are visualized with mean and standard deviation computed from three random seeds each. We compare DF-ExpEnse against two baselines: vanilla DSRL and Max-Q, which selects the action candidate that has the highest estimated value according to the critic ensemble. This exploitative technique has been explored in prior works such as EMaQ (Ghasemipour et al., 2021) and Q-Chunking (Li et al., 2026), and is equivalent to DF-ExpEnse when $\alpha = 0$. As with DF-ExpEnse, each action candidate's value is estimated as the minimum prediction from the ensemble to avoid overestimation. As only the Q-value is considered during action selection applying fleet normalization is superfluous for Max-Q, and does not affect each agent's choice.

We also compare the DSRL-integrated version of DF-ExpEnse against two alternative strategies for evaluating action exploration quality: Exploration via Elliptical Episodic Bonuses (E3B) (Henaff et al., 2022) and Plan2Explore (Sekar et al., 2020). Neither approach has previously been applied to online reinforcement learning finetuning of pretrained behavior cloned policies; we thus apply the necessary adaptations to ensure fair comparison.

Whereas vanilla E3B annotates collected trajectories collected by the policy with bonus rewards, here we implement E3B to perform action candidate selection for improved online exploration collection. This online version of E3B uses a learned transition dynamics model to predict the future states associated with each action candidate; the exploration interest is then computed as the value estimate from the critic ensemble augmented with an Elliptical Episodic Bonus term. As the E3B authors note that the choice of continuous state representation is important, we benchmark against two versions of state encoding: learned representations of robotic proprioception (E3B-State), and a pretrained DINO representation of the pixel observation (E3B-DINO). The action candidate set size is kept uniform with default DF-ExpEnse for fairness, and all underlying reinforcement learning finetuning settings are maintained for consistency.

For Plan2Explore an ensemble of one-step transition dynamics models is used, as in the original work, to evaluate action candidates. Exploration interest for each candidate is estimated through the variance of predicted future states across the dynamics ensemble; the action that produces the state with the highest approximated variance is selected for execution. As with E3B, candidate set size and underlying DSRL finetuning settings are kept consistent with DF-ExpEnse.

Lastly, E3B and Plan2Explore both use additional model components: a transition dynamics model, and an ensemble of transition dynamics models, respectively, to evaluate actions during online experience collection. In contrast, DF-ExpEnse naturally reuses the critic ensemble needed for policy gradient optimization in an online manner, and does not require extra modules to perform principled exploration.

**RoboMimic Results:** We present RoboMimic performance in the first row of Figure 2. We first observe that DF-ExpEnse achieves consistent sample efficiency improvements, demonstrating a higher success rate than alternative methods given the same amount of observed timesteps from the environment. We first note that on Lift, all methods achieve comparable finetuning performance; we attribute this to the simplicity of the task, being the easiest in the

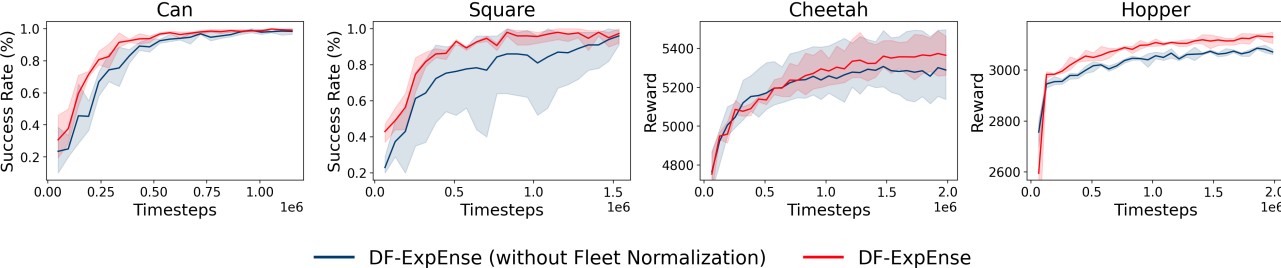

*Figure 5.* **Fleet Normalization Ablation**. We demonstrate across both RoboMimic and OpenAI Gym tasks that fleet normalization is a meaningful component of DF-ExpEnse, and that removing it impacts sample efficiency. All results are averages over three random seeds.

suite, as well as the high initial pretrained policy performance. For the more challenging RoboMimic tasks where base performance is lower and exploration over behaviors during experience collection may be more productive, DF-ExpEnse significantly improves over the baselines. Of note is that Max-Q underperforms against even vanilla DSRL across all the manipulation tasks. We hypothesize this arises due to Max-Q being naturally exploitative behavior, which may reinforce erroneous prior beliefs over behavior without exploration. The modes may also converge on a suboptimal action mode that satisfies an underexplored critic ensemble, causing both quality and diversity of sampled action candidates to collapse. On the other hand, DF-ExpEnse includes an additional critic disagreement term, and utilizes BC-SR to regularize the action candidate space to avoid pitfalls encountered by Max-Q. We verify that each of these design decisions indeed meaningfully improves exploration and sample-efficiency during finetuning in our ablations study.

As observed in Figure 2, DF-ExpEnse also consistently outperforms alternative strategies for evaluating action exploration quality. State-based E3B performs the same as or worse than DINO-based E3B for all tasks. As E3B is known to be sensitive to the state encoding used, we hypothesize this may be because large-scale pretraining priors from DINO can be useful for certain robotic control tasks that feature more realistic visual settings. For example, the difference between E3B-State and E3B-DINO is largest for RoboMimic Lift and Can, which feature common objects such as a cube or a can; however, the difference between the two is not pronounced in environments with more synthetic visuals such as OpenAI Gym tasks. Meanwhile, DF-ExpEnse behavior does not depend on the quality of any state embeddings when performing principled exploration.

Whereas DF-ExpEnse outperforms baselines on Tool Hang and Lift, the respective difficulty and simplicity of these tasks makes them difficult to convey the effect of adjustments to the method. Alternatively, both Can and Square clearly demonstrate varied performance trends across different methods; we thus utilize these tasks for subsequent ablations to illustrate design decisions relevant to DF-ExpEnse.

**OpenAI Gym Results:** In the second row of Figure 2 we observe superior reward curve comparisons for DF-ExpEnse against the baselines, supporting our hypothesis that improving exploration quality during online experience collection can directly lead to increased finetuning sample-efficiency.

**DexMimicGen Results:** In Figure 3 we observe superior success-rate curve comparisons for DF-ExpEnse against the ResFiT baseline it was built on top of as well as a Max-Q action selection scheme. We thus show that DF-ExpEnse is a general and versatile exploration technique that can be seamlessly integrated with a variety of finetuning strategies.

### 4.6. Behavior Cloning Sampling Regularization Matters

A key component of DF-ExpEnse is its treatment of the multimodal modeling capability of the diffusion policy as an effective way to filter the continuous action space. From this perspective, we hope that samples proposed by the agent are not only of reasonable base quality, but also exhibit broad coverage of the action space. As a policy may naturally begin to converge on fewer action modes throughout finetuning, BC-SR is used to regularize the candidate set to include samples from the initial pretrained policy, which intuitively may preserve more multimodality than a policy that has further finetuned on its own online experience. We thus empirically investigate the exploration benefits of BC-SR.

In Figure 4, along with existing comparisons between Max-Q and DF-ExpEnse, we report two additional settings: adding BC-SR to Max-Q and removing BC-SR from DF-ExpEnse. We discover that across all tasks, the two ablation experiments lie between the performance curves of Max-Q and DF-ExpEnse; in essence, removing BC-SR from DF-ExpEnse consistently hurts its performance, while adding BC-SR to Max-Q consistently improves it. We attribute the dramatic performance upgrade of Max-Q with BC-SR over standard Max-Q to BC-SR's ability to mitigate the tendency Max-Q may have to rapidly collapse to a suboptimal mode given its exploitative behavior. This reinforces the important role that multimodality has in filtering the continuous action space to a set of action candidates, and highlights BC-SR as a generally helpful component during online inference.

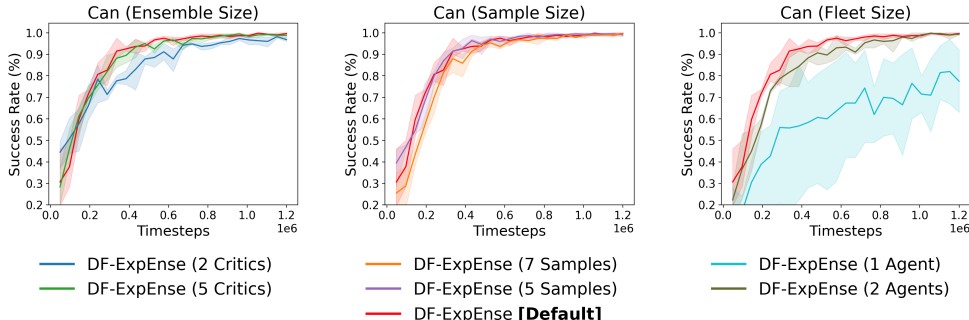

*Figure 6.* **Studying Component Sizes**. We ablate the critic ensemble size, action candidate sample size, and fleet size for the RoboMimic Can task. We find that DF-ExpEnse benefits from a larger ensemble and fleet size, and exhibits saturation with increases to the sample set.

Furthermore, as using Max-Q with BC-SR improves but does not completely reach the performance of DF-ExpEnse across all experiments, this suggests that other components of DF-ExpEnse also meaningfully contribute to improved exploration. These includes scoring actions using an additional critic disagreement term, and performing fleet normalization to contextualize action candidates across the group.

### 4.7. Fleet Normalization Matters

Cross-fleet collaboration for exploration is an underexplored but promising direction towards sample-efficient finetuning, particularly as parallelization efforts improve and real-world robotic deployments increase. In fleet settings, DF-ExpEnse enables agents to communicate statistics regarding their individual action candidates with the rest of the group before action selection. Each agent then contextualizes the value and uncertainty estimates of their personal choices against the situations of all other agents, thus facilitating collaboration across the fleet for improved group-aware exploration.

In Figure 5, we compare DF-ExpEnse with fleet normalization against a same-sized fleet where each parallelized DF-ExpEnse agent performs online experience collection in isolation. We verify that fleet normalization consistently improves the quality of experience collection as a group, facilitating increased sample efficiency across all evaluations.

### 4.8. Impact of Candidate Set Size, Ensemble Size, and Fleet Size

We ablate size hyperparameters for the three main components that describe DF-ExpEnse's behavior: the critic ensemble, the action candidate set, and the fleet. We report comparisons on the RoboMimic Can task against the default DF-ExpEnse setting in Figure 6, averaged over three seeds.

In the first plot, we find that a larger critic ensemble helps performance; using 2 critics results in significantly decreased sample efficiency performance. Performance appears to saturate with 5 critics, which achieves comparable performance with using the default size of 10. We believe

that too small an ensemble may result in a noisier estimate of the uncertainty the agent has in a given candidate action, leading to a more random selection from the candidate set.

In the second plot, we compare using larger action candidate sample sizes of 5 and 7 against the default setting of 3. In all cases, we continue to apply BC-SR with a regularization size of 1. This setting evaluates to what effect seeing a larger amount of samples from the policy has on exploration. We observe that whereas there are slight improvements in utilizing a sample size of 5, the sample efficiency is ultimately comparable across all settings. Intuitively, we expect saturation in performance gains as we scale up the sample size past a certain extent; at sufficiently large sample sizes we expect samples to begin to represent redundant modes.

In the third plot, we compare the effect that fleet size has on DF-ExpEnse performance. Intuitively, larger fleets may provide greater amounts of normalization and collaboration possibilities. We indeed observe that performance decreases with smaller fleet sizes, down to no effective fleet when only a single agent is available. Extended comparisons across fleet size settings in Appendix B also express similar trends.

## 5. Conclusion

We present DF-ExpEnse, an exploration technique that improves sample efficiency when finetuning a pretrained generative control policy via reinforcement learning. DF-ExpEnse evaluates the exploration interest of an action by using an ensemble of critics to estimate its value and uncertainty. To tractably identify an action that balances quality with uncertainty, DF-ExpEnse uses the multimodal modeling capability of the generative control policy to produce a set of reasonable candidates. Furthermore, in fleet settings, DF-ExpEnse performs cross-fleet communication to better contextualize action selection for each agent and facilitate collaborative exploration. DF-ExpEnse can be flexibly integrated with multiple reinforcement learning techniques that finetune generative policies; experimentally, evaluated over a variety of manipulation and locomotion tasks, we find that it consistently improves online finetuning sample-efficiency.

## Acknowledgments

This work was done when Calvin was visiting the REALab at Stanford University, supported by the Research Mobility Fellowship from Brown University. We would like to thank Zhanyi Sun for valuable advice and technical support at notable stages of the project, Yiding Jiang for helpful discussions related to exploration via ensembles, and Zilai Zeng and Zitian Tang for their timely help with experimental runs. This material is partially supported by NSF IIS-2433429 and IIS-2543166, a Seed Award from Brown University, and the NVIDIA Academic Grant Program. Our research was conducted using computational resources at the Center for Computation and Visualization at Brown University.

## Impact Statement

This paper seeks to increase the sample efficiency of finetuning pretrained diffusion policies through reinforcement learning by designing an improved exploration strategy. As robotic deployments increase in the real world, this work can help to facilitate faster skill learning or refinement. At the same time this work, like many other reinforcement learning works, makes minimal assumptions about the task being optimized for, which could be a negative behavior when used by bad actors. We do not feel there are pressing societal consequences to highlight regarding our work.

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

## A. All Base Policy and DSRL Training Hyperparameters

We provide common hyperparameter settings used in our DSRL-integrated DF-ExpEnse experiments in Tables 1, 2, and 3. When DF-ExpEnse is applied on top of DSRL as the underlying reinforcement learning algorithm of choice, it is only utilized during the online experience collection step. As such, many of the optimization hyperparameters of DSRL are untouched, to maintain base performance and attribute any performance improvements directly to the quality of experience collected through DF-ExpEnse. Of these tasks, Tool Hang is entirely new, and we thus propose the entire suite of task-related hyperparameters. Lastly, in the spirit of evaluating true sample efficiency, we do not collect any initial steps across any tasks or environments for both DSRL and DF-ExpEnse.

*Table 1.* **Hyperparameters for DSRL Robomimic experiments.** Hyperparameters that deviate from the base DSRL implementation are reported with an asterisk (\*).

| Hyperparameter | Lift | Can | Square | Tool Hang$^*$ |
|---|---|---|---|---|
| Action chunk size | 4 | 4 | 4 | $8^*$ |
| Hidden size | 2048 | 2048 | 2048 | $2048^*$ |
| Gradient steps per update | 30 | 20 | 20 | $20^*$ |
| $Q^{\mathcal{W}}$ update steps | 10 | 10 | 10 | $10^*$ |
| Discount factor | 0.99 | 0.99 | 0.999 | $0.99^*$ |
| Action magnitude ($b_{\mathcal{W}}$) | 1.5 | 1.5 | 1.5 | $1^*$ |
| Initial steps | $0^*$ | $0^*$ | $0^*$ | $0^*$ |
| Critic Ensemble Size | $10^*$ | $10^*$ | $10^*$ | $10^*$ |
| $\pi_{\mathrm{dp}}$ train denoising steps | 20 | 20 | 100 | $10^*$ |
| $\pi_{\mathrm{dp}}$ inference denoising steps | 8 | 8 | 8 | $10^*$ |

*Table 2.* **Hyperparameters for DSRL OpenAI Gym experiments.** Hyperparameters that deviate from the base DSRL implementation are reported with an asterisk (\*).

| Hyperparameter | Hopper-v2 | Walker2D-v2 | HalfCheetah-v2 |
|---|---|---|---|
| Action chunk size | 4 | 4 | 4 |
| Hidden size | 2048 | 2048 | 1024 |
| Gradient steps per update | 20 | 20 | 20 |
| $Q^{\mathcal{W}}$ update steps | 10 | 10 | 10 |
| Discount factor | 0.99 | 0.99 | 0.99 |
| Action magnitude ($b_{\mathcal{W}}$) | 1.5 | 2.5 | 1.5 |
| Initial steps | $0^*$ | $0^*$ | $0^*$ |
| Critic Ensemble Size | $10^*$ | 2 | $10^*$ |
| $\pi_{\mathrm{dp}}$ train denoising steps | 20 | 20 | 20 |
| $\pi_{\mathrm{dp}}$ inference denoising steps | 5 | 5 | 5 |

*Table 3.* **Common DSRL hyperparameters for online experiments.** We do not adjust any base hyperparameters, and report the default common DSRL hyperparameters below.

| Hyperparameter | Value |
|---|---|
| Learning rate | 0.0003 |
| Batch size | 256 |
| Activation | Tanh |
| Target entropy | 0 |
| Target update rate ($\tau$) | 0.005 |
| Number of actor and critic layers | 3 |
| Fleet size | 4 |

## B. Baseline Comparisons at Varying Fleet Sizes

To continue the investigation on fleet size from Section 4.8 and Figure 6, we report and compare the performance of baseline methods for both larger and smaller fleet sizes. Firstly we find that below a fleet size of 4, DF-ExpEnse performance does decrease, verifying that DF-ExpEnse can leverage larger fleet sizes to help improve sample efficiency. Nevertheless, DF-ExpEnse still reliably outperforms vanilla DSRL and Max-Q across all fleet sizes, large and small. Interestingly, with a fleet size of 16, the Max-Q selection algorithm completely collapses across all three seeds. These findings further reinforce DF-ExpEnse as a robust method that can be applied on top of standard reinforcement learning finetuning techniques to provide consistent sample efficiency benefits across available resource settings. Furthermore, we highlight the superiority of the design decisions behind DF-ExpEnse, in comparison to alternative action selection schemes such as Max-Q.

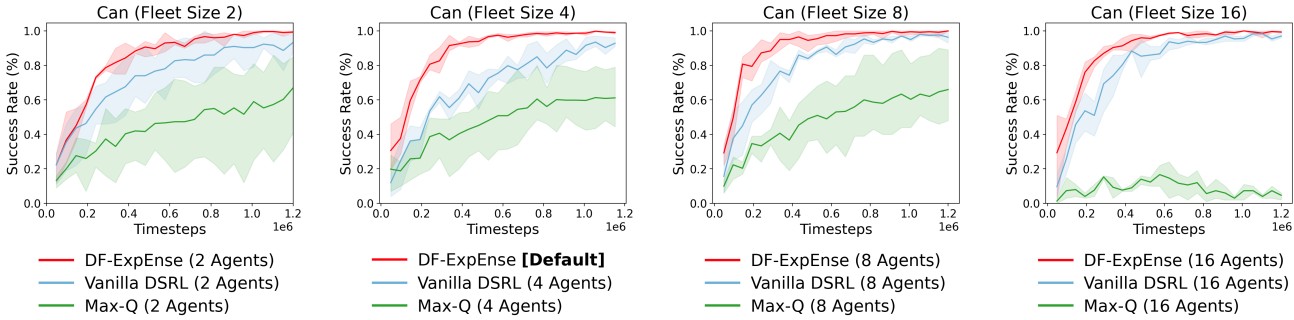

*Figure 7.* **Comparing Varying Fleet Sizes**. We ablate fleet sizes on the Can task, and report baseline comparisons over three random seeds. DF-ExpEnse consistently achieves superior sample efficiency, demonstrating its versatility across available resource settings.

## C. Critic Ensemble Size for Walker

We note that the results for Walker reported in Figure 2 feature a critic ensemble of size 2, as also listed in Table 2, which aligns with default DSRL settings. Whereas scaling up the ensemble size to 10, as is used in all other tasks, still resulted in DF-ExpEnse achieving and maintaining a high reward for Walker, we found that vanilla DSRL with a critic ensemble of size 10 seemed to collapse when reusing the rest of the DSRL hyperparameters settings. In Section 4.5 and Figure 2 we thus showcase DF-ExpEnse on a hyperparameter setting that is selected to be amenable to the baseline; nevertheless, we still observe consistent improvement gains arise from our method. However, in order to maintain consistency across tasks when examining DF-ExpEnse settings themselves, we focus on Cheetah and Hopper when performing subsequent ablative studies.

## D. Default UCB Value Estimation through Ensemble Means

In evaluating the exploration interest of each action candidate, we compute the value estimate using the minimum of the ensemble (Equation 2) to avoid potential value overestimation. However, in the standard UCB approach, a mean over the ensemble is used. We perform additional ablations to experimentally investigate if this design decision impacts downstream performance. In Figure 8, across both Robomimic and OpenAI Gym tasks, we find that using the mean achieves comparable performance with using the ensemble minimum. DF-ExpEnse is therefore robust to this particular design decision.

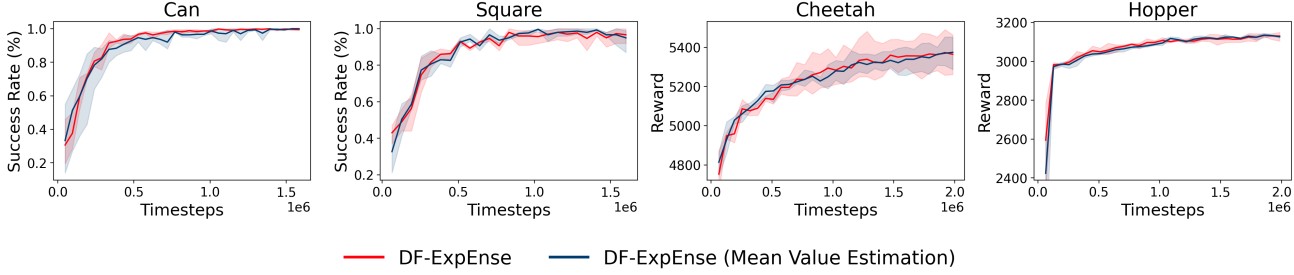

*Figure 8.* **Ensemble Means for Value Estimation**. We investigate performance across Can and Square tasks from Robomimic and Cheetah and Hopper tasks from OpenAI Gym when using mean value estimation, averaged over three random seeds. We empirically discover that there are no substantial differences from using the ensemble minimum; DF-ExpEnse is thus robust to this design decision.

# E. Exploration-Weighted Sampling for Action Selection

By default, action selection in DF-ExpEnse is performed by selecting the candidate with the maximum estimated exploration interest. However, we can consider an alternative procedure which first constructs a discrete distribution over the candidates weighted by their estimated exploration interest terms, and then selects the action to execute via sampling. This is similar to the selection strategy performed in V-GPS (Nakamoto et al., 2024); however, whereas V-GPS scores actions according to value estimates alone, in this work actions are evaluated with respect to exploration interest to improve online experience collection quality. We report our findings across multiple tasks in Figure 9, and discover that exploration-weighted sampling for action selection achieves comparable performance to maximum selection. However, in the Square task, we find slight divergence in eventual performance after extended timesteps; we thus default DF-ExpEnse to utilize the maximum exploration interest selection scheme, as sampling can require additional parameters to tune such as temperature weighting and schedules to make stable. Nevertheless, DF-ExpEnse does naturally support sampling-based action selection strategies.

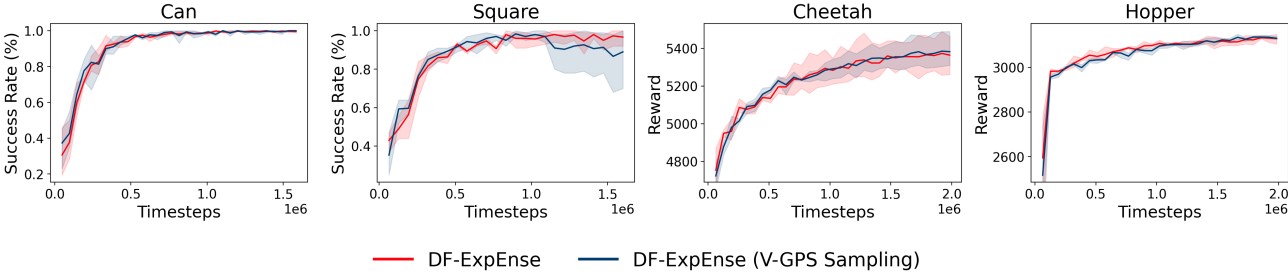

*Figure 9.* **Exploration-Weighted Sampling for Action Selection**. We investigate performance across Robomimic and OpenAI Gym tasks when using V-GPS style sampling for action selection, averaged over three random seeds. Whereas there is largely no substantial differences from default DF-ExpEnse during initial optimization, performance for Square does appear to eventually exhibit some divergence at extended timesteps. Adjusting parameters relevant to the sampling procedure can potentially mitigate this phenomenon.

# F. Future Work and Limitations

Currently, DF-ExpEnse performs exploration at every single timestep; however, exploration may be less necessary in certain states or given the agent's currently mastered behaviors. For example, in a robotic insertion task, an agent may already have learned to pick up the object and move it towards the insertion goal proficiently. Then, intuitively the agent should focus on exploring around the insertion states, rather than commit further computation resources to exploring during the mastered pick-and-move behavior. In such scenarios, indiscriminately performing exploration at every timestep indiscriminately may be wasteful or even suboptimal compared to direct inference. Dynamically determining when to perform an exploration step in DF-ExpEnse, conditioned on an agent's current state and learned behaviors, is thus interesting future work to explore.

Furthermore, whereas fleet normalization provides performance benefits, in real world deployments it may be impractical to have all agents communicate with each other due to latency and distance considerations. However, we notice in the ablations provided in Section B and Figure 7 that scaling fleet size may eventually encounter saturation in performance benefits, as there are no further substantial gains when scaling beyond a fleet size of 4 on the Can task. Thus, we can potentially split large-scale deployments into smaller clusters, and then perform fleet normalization locally within them. Investigating the effect of sharding parallel environments combined with localized fleet normalization is interesting to pursue as future work.

# G. DF-ExpEnse Pseudocode (Noise Input Optimization)

---

**Algorithm 1** Diffusion-Filtered Exploration via Ensembles (DSRL-NA Integration)

---

**Input:** Pretrained Diffusion Policy $\pi_{\text{dp}}$, Sample Size $M$, Online Environments $\mathcal{M}_1, ..., \mathcal{M}_N$
**Initialize:** Noise Policy $\pi_\theta$, Action Critic Ensemble $Q^{\text{action}}$ : $[Q_{\phi_1}, \ldots Q_{\phi_K}]$, Noise Critic Ensemble $Q^{\text{noise}}$ : $[Q_{\psi_1}, \ldots Q_{\psi_K}]$, Empty Experience Buffer $\mathfrak{B}$, Empty Action Buffer $\mathcal{A}$

1: **repeat**
2:     ▷ *Generate and Evaluate Action Candidates Across Fleet*
3:     $\mathcal{A} = \{\}$
4:     **for** $m = 1, \ldots, M$ **do**
5:        **for** $n = 1, \ldots, N$ **do**
6:           $\boldsymbol{a}^{\text{noise}}_{m,n,t} \sim \pi_\theta(\boldsymbol{s}_{n,t})$
7:           $\boldsymbol{a}_{m,n,t} \sim \pi_{\text{dp}}(\boldsymbol{s}_{n,t}, \boldsymbol{a}^{\text{noise}}_{m,n,t})$
8:           $\mathcal{A} \cup \{\boldsymbol{a}_{m,n,t}\}$
9:           $v_{n,m} = \min\left(Q(\boldsymbol{a}_{m,n,t}, \boldsymbol{s}_{n,t})\right)$
10:          $d_{n,m} = \text{std}\left(Q(\boldsymbol{a}_{m,n,t}, \boldsymbol{s}_{n,t})\right)$
11:        **end for**
12:     **end for**
13:     ▷ *Compute Fleet Normalization*
14:     $\bar{v}_{n,m} = \frac{v_{n,m} - \text{avg}([v_{1,1}, \ldots, v_{N,M}])}{\text{std}([v_{1,1}, \ldots, v_{N,M}])}$
15:     $\bar{d}_{n,m} = \frac{d_{n,m} - \text{avg}([d_{1,1}, \ldots, d_{N,M}])}{\text{std}([d_{1,1}, \ldots, d_{N,M}])}$
16:     $\bar{e}_{n,m} = \bar{v}_{n,m} + \alpha * \bar{d}_{n,m}$
17:     ▷ *Select Action and Execute*
18:     **for** $n = 1, \ldots, N$ **do**
19:        $\boldsymbol{a}_{m^\star,n,t} \leftarrow \underset{\boldsymbol{a} \in \mathcal{A}}{\arg\max} \; [\bar{e}_{1,n}, \ldots, \bar{e}_{M,n}]$
20:        Step environment $\mathcal{M}_n$ with $\boldsymbol{a}_{m^\star,n,t}$ and observe reward $r_{n,t}$ and next state $\boldsymbol{s}_{n,t+1}$
21:        Add $(\boldsymbol{s}_{n,t}, \boldsymbol{a}^{\text{dp}}_{m^\star,n,t}, \boldsymbol{a}_{m^\star,n,t}, r_{n,t}, \boldsymbol{s}_{n,t+1})$ to $\mathfrak{B}$
22:     **end for**
23:     ▷ *Update Action Critic Ensemble, Noise Critic Ensemble, and Noise Policy*
24:     Update $Q_{\phi_i}$: $\underset{\phi_i}{\min} \; \mathbb{E}_{(\boldsymbol{s},\boldsymbol{a},r,\boldsymbol{s}') \sim \mathfrak{B}, \boldsymbol{a}' \sim \pi_{\text{dp}}(\boldsymbol{s}', \pi_\theta(\boldsymbol{s}'))} \left[(Q_{\phi_i}(\boldsymbol{s}, \boldsymbol{a}) - r - \gamma * \min Q^{\text{action}}(\boldsymbol{s}', \boldsymbol{a}'))^2\right]$
25:     Update $Q_{\psi_i}$: $\underset{\psi_i}{\min} \; \mathbb{E}_{\boldsymbol{s} \sim \mathfrak{B}, \boldsymbol{\epsilon} \sim \mathcal{N}(0,I)} \left[\left(Q_{\psi_i}(\boldsymbol{s}, \boldsymbol{\epsilon}) - Q_{\phi_i}\left(\boldsymbol{s}, \pi_{\text{dp}}(\boldsymbol{s}, \boldsymbol{\epsilon})\right)\right)^2\right]$
26:     Update $\pi_\theta$: $\underset{\theta}{\max} \; \mathbb{E}_{\boldsymbol{s} \sim \mathfrak{B}} \left[\min Q^{noise}(\boldsymbol{s}, \pi_\theta(\boldsymbol{s}))\right]$
27: **until** convergence

---

# H. DF-ExpEnse Pseudocode (Residual Optimization)

---

**Algorithm 2** Diffusion-Filtered Exploration via Ensembles (ResFiT Integration)

---

**Input:** Pretrained Diffusion Policy $\pi_{\text{dp}}$, Sample Size $M$, Online Environments $\mathcal{M}_1, ..., \mathcal{M}_N$

**Initialize:** Residual Policy $\pi_\theta$, Critic Ensemble $Q_{\phi_1}, \ldots Q_{\phi_K}$, Empty Experience Buffer $\mathfrak{B}$, Empty Action Buffer $\mathcal{A}$

1: **repeat**
2:     ▷ *Generate and Evaluate Action Candidates Across Fleet*
3:     $\mathcal{A} = \{\}$
4:     **for** $m = 1, \ldots, M$ **do**
5:         **for** $n = 1, \ldots, N$ **do**
6:             $\boldsymbol{a}_{m,n,t}^{\text{dp}} \sim \pi_{\text{dp}}(\boldsymbol{s}_{n,t})$
7:             $\boldsymbol{a}_{m,n,t}^{\text{res}} \sim \pi_\theta(\boldsymbol{s}_{n,t}, \boldsymbol{a}_{m,n,t}^{\text{dp}})$
8:             $\boldsymbol{a}_{m,n,t} = \boldsymbol{a}_{m,n,t}^{\text{dp}} + \boldsymbol{a}_{m,n,t}^{\text{res}}$
9:             $\mathcal{A} \cup \{\boldsymbol{a}_{m,n,t}\}$
10:            $v_{n,m} = \min\left(Q(\boldsymbol{a}_{m,n,t}, \boldsymbol{s}_{n,t})\right)$
11:            $d_{n,m} = \text{std}\left(Q(\boldsymbol{a}_{m,n,t}, \boldsymbol{s}_{n,t})\right)$
12:         **end for**
13:     **end for**
14:     ▷ *Compute Fleet Normalization*
15:     $\bar{v}_{n,m} = \frac{v_{n,m} - \text{avg}([v_{1,1}, \ldots, v_{N,M}])}{\text{std}([v_{1,1}, \ldots, v_{N,M}])}$
16:     $\bar{d}_{n,m} = \frac{d_{n,m} - \text{avg}([d_{1,1}, \ldots, d_{N,M}])}{\text{std}([d_{1,1}, \ldots, d_{N,M}])}$
17:     $\bar{e}_{n,m} = \bar{v}_{n,m} + \alpha * \bar{d}_{n,m}$
18:     ▷ *Select Action and Execute*
19:     **for** $n = 1, \ldots, N$ **do**
20:         $\boldsymbol{a}_{m^\star,n,t} \leftarrow \underset{\boldsymbol{a} \in \mathcal{A}}{\arg\max}\ [\bar{e}_{1,n}, \ldots, \bar{e}_{M,n}]$
21:         Step environment $\mathcal{M}_n$ with $\boldsymbol{a}_{m^\star,n,t}$ and observe reward $r_{n,t}$ and next state $\boldsymbol{s}_{n,t+1}$
22:         Add $(\boldsymbol{s}_{n,t}, \boldsymbol{a}_{m^\star,n,t}^{\text{dp}}, \boldsymbol{a}_{m^\star,n,t}, r_{n,t}, \boldsymbol{s}_{n,t+1})$ to $\mathfrak{B}$
23:     **end for**
24:     ▷ *Update Critic Ensemble and Residual Policy*
25:     Update $Q_{\phi_i}$: $\underset{\phi_i}{\min}\ \mathbb{E}_{(\boldsymbol{s}, \boldsymbol{a}, r, \boldsymbol{s}') \sim \mathfrak{B},\, \boldsymbol{a}^{\text{dp}'} \sim \pi_{\text{dp}}(\boldsymbol{s}')} \left[(Q_{\phi_i}(\boldsymbol{s}, \boldsymbol{a}) - r - \gamma * \min Q(\boldsymbol{s}', \boldsymbol{a}^{\text{dp}'} + \pi_\theta(\boldsymbol{s}', \boldsymbol{a}^{\text{dp}'})))^2\right]$
26:     Update $\pi_\theta$: $\underset{\theta}{\max}\ \mathbb{E}_{(\boldsymbol{s}, \boldsymbol{a}^{dp}) \sim \mathfrak{B}, i}\left[Q_{\phi_i}(\boldsymbol{s}, \boldsymbol{a}^{dp} + \pi_\theta(\boldsymbol{s}, \boldsymbol{a}^{dp}))\right]$
27: **until** convergence

---

