# OpenReview forum: "DF-ExpEnse: Diffusion Filtered Exploration for Sample Efficient Finetuning"
_ICML.cc/2026/Conference — ICML 2026 regular_

### Official Review · Reviewer_yojD · 2026-03-12

**Soundness:** 3
**Presentation:** 3
**Significance:** 2
**Originality:** 3
**Overall Recommendation:** 4
**Confidence:** 3

**Summary:**

This paper proposes DF-ExpEnse (Diffusion Filtered Exploration via Ensembles), a method aimed at improving the sample efficiency of RL finetuning for pretrained generative control policies, particularly diffusion-based policies used in robotics.

**Compliance With Llm Reviewing Policy:**

Affirmed.

**Key Questions For Authors:**

+ How sensitive is the method to quality of pretrained diffusion policy? If the diffusion policy is poorly trained or trained on biased data, could the filtering step prevent exploration of better actions outside the prior distribution?
+ Since exploration interest depends heavily on the critic ensemble, how does the method perform in the very early stages of finetuning if the critics are poorly initialized?

**Limitations:**

yes

**Strengths And Weaknesses:**

**Strength**
+ This paper focuses on an important problem of sample-efficient online finetuning of pretrained policies.
+ The idea of combining diffusion-generated candidate actions with ensemble-based exploration is intuitive. This work integrates naturally with diffusion policies.
+ The experimental results demonstrate improvements in performance compared to baselines. The authors report that the proposed exploration strategy is beneficial.

**Weakness**
+ The method introduces additional computation due to sampling multiple actions from the diffusion model and evaluating each action with a Q-ensemble. However, the paper does not thoroughly analyze the computational cost.
+ The paper compares the proposed method with only two baselines. Including additional SOTA methods would make the experimental results more convincing.

---

> ### Author Rebuttal · Authors · 2026-03-31
>
> We thank Reviewer yojD for their comments and feedback on our work; we are happy to hear that the reviewer appreciates the intuitive nature of our approach as well as the importance of the problem we seek to address. We seek to address their questions below:
>
> **On Computational Cost:** We compute and provide wall clock time comparisons when using DF-ExpEnse below on the Robomimic Can task, ablating across ensemble size and action candidate sample size, and fleet size:
>
> |Ensemble Size|Sample Size|Fleet Size|Average Action Selection Time (ms)|
> |-|-|-|-|
> |**10**|**3**|**4**|19.004|
> |10|3|8|19.120|
> |10|3|16|19.226|
> |10|5|4|19.328|
> |10|7|4|19.504|
> |5|3|4|14.285|
> |2|3|4|10.614|
>
> The first bold row represents the default settings.  All timing numbers are computed as averages over 100 action generation-and-selection steps.  Scaling the sample size is not as impactful on action selection timing as scaling the critic ensemble size for evaluating exploration interest.
>
> Simultaneously, we note that much of DF-ExpEnse is parallelizable: action candidate generation and initial critic ensemble evaluation can all be performed in parallel to each other, and thus wall clock overhead can be mitigated with additional engineering and resources. Furthermore, DF-ExpEnse outperforms in sample-efficient finetuning performance against alternate approaches that incur the same additional computational cost (e.g. Max-Q, Figures 2 and 3). This suggests that DF-ExpEnse is the preferable approach when considering how computation resource scale should be utilized for improving online finetuning performance.
>
> **Additional Baselines:** We consider an additional performant approach for finetuning pretrained BC policies through online reinforcement learning: Residual RL Finetuning. We integrate DF-ExpEnse on top of ResFiT [2], which extends ResiP [3].  Furthermore, we evaluate DF-ExpEnse on three tasks from a novel environment suite: DexMimicGen [1], which features complex dextrous manipulation tasks with sparse rewards. We provide the comparison plots here ([Comparison Figure](https://sites.google.com/view/dfexpense-rebuttal-figures/#h.8b2c1gau74nv)), where we find that DF-ExpEnse consistently facilitates superior sample efficiency in online RL finetuning. This reinforces DF-ExpEnse as a general technique that improves exploration quality and sample-efficient finetuning across multiple performant RL finetuning approaches, and across multiple tasks and environment settings.
>
> **Sensitivity to Pretrained BC Quality:** We agree that this is an interesting question to consider. RL finetuning of early checkpoints, RL finetuning quality may suffer because the BC policy may have less accurate modeling of reasonable actions. On the other hand, extended pretraining may eventually overfit to the training set and exhibit less sample diversity, utilizing such BC policy checkpoints may lead to lower exploration quality. Whereas our method focuses on a general exploration strategy that can be applied irrespective of checkpoint quality, we provide experimental evaluations ([Comparison Figure](https://sites.google.com/view/dfexpense-rebuttal-figures/#h.lz82ebotrikf)) comparing the effect of BC pretraining on RL finetuning through DF-ExpEnse, on the RoboMimic Can task where each run is averaged over three seeds. We verify that early checkpoints struggle in terms of sample efficient self-improvement compared to later checkpoints. For reference, vanilla DF-ExpEnse was integrated on top of a DSRL implementation that utilized a 5000-epoch pretrained BC policy. Checkpoints that observed 1000 or 2000 epochs during pretraining appear to achieve a slight average increase in sample-efficiency, suggesting that reasonably early checkpoints that best balance modeling quality and sample diversity may have potential synergy with DF-ExpEnse in exploration and sample-efficient self-improvement.
>
> **Critic Initialization:** We clarify that the critics are already randomly initialized; the early steps reported in all graphs indeed correspond to poor initialization at early stages of finetuning. Nonetheless, we find that DF-ExpEnse can still achieve faster performance improvements compared to alternate approaches. As the critic ensemble is now used both in action selection as well as policy gradient updates, we believe the online experience collected through DF-ExpEnse is helpful for sample efficient critic improvement as well, even from random initialization. We believe that better critic ensemble initialization can help improve DF-ExpEnse performance further; though we leave these specific design decisions as future work beyond the current most general approach.
>
> [1] Jiang et al., DexMimicGen: Automated Data Generation for Bimanual Dexterous Manipulation via Imitation Learning, ICRA 2025.
>
> [2] Ankile et al., Residual Off-Policy RL for Finetuning Behavior Cloning Policies, arXiv 2025.
>
> [3] Ankile et al., From Imitation to Refinement - Residual RL for Precise Assembly, ICRA 2025.

---

> > ### Author Rebuttal · Reviewer_yojD · 2026-04-05
> >
> > Thanks. I will keep my score.

---

> > > ### Author Response · Authors · 2026-04-05
> > >
> > > We appreciate the reviewer’s time and consideration.  To the reviewer’s two listed weaknesses, we have provided additional thorough analysis on computational cost and additional experimental results for DF-ExpEnse integrated with Residual RL finetuning (a separate performant strategy for RL finetuning of pretrained BC policies).  We also provide additional baseline comparisons against alternate strategies for evaluating action exploration quality, where we found that DF-ExpEnse continues to outperform other approaches while not requiring additional model components to conduct exploration (unlike E3B and Plan2Explore) ([Comparison Figure](https://sites.google.com/view/dfexpense-rebuttal-figures/#h.8ol5m1oy014c)).  To the reviewer’s two listed questions, we have provided additional experimental ablations on the effect of checkpoint quality, as well as clarification on critic initialization.  We thank the reviewer for the helpful feedback - all of these additional experiments, ablations, and analyses improve our manuscript, and we will incorporate them in the final draft.
> > >
> > > We notice that the reviewer has selected the acknowledgment option of “Partially resolved - I have follow-up questions for the authors”.  We welcome any elaboration on such further questions so that we may directly address them in the remainder of the discussion period.

---

### Official Review · Reviewer_Kf7a · 2026-03-13

**Soundness:** 2
**Presentation:** 4
**Significance:** 3
**Originality:** 3
**Overall Recommendation:** 4
**Confidence:** 4

**Summary:**

The paper presents Diffusion filtered exploration via Ensembles (DF-ExpEnse) to increase the sampling efficiency of the finetuning procedure. DF-ExpEnse learns a highly expressive candidate set that is then selected by an ensemble of critics to identify actions of high exploration interest. In a parallelized fleet, DF-ExpEnse facilitates group exploration through cross-agent communication. The paper presents a behavior cloning regularization technique that maintains the multimodal prior of the offline data during online inference. The action selection in the end happens by taking an argmax over the fleet-normalized exploration interest estimates. The method is evaluated on locomotion and manipulation benchmarks.

**Compliance With Llm Reviewing Policy:**

Affirmed.

**Final Justification:**

The rebuttal helped

**Key Questions For Authors:**

Q1. How many denoising steps are you using?

Q2. What is the wall clock time of each step?

Q3. What is the intra-fleet communication overhead?

Q4. Why are you not using a larger fleet?

Q5. How does your work compare to Flow Q learning?
Park, Seohong et al. “Flow Q-Learning.” ArXiv abs/2502.02538 (2025): n. pag.

If the experimental concerns are addressed in tandem with the necessary related work comparisons, I am happy to increase my score.

**Limitations:**

yes

**Strengths And Weaknesses:**

1. The paper is easy to follow

2. The figures are clear and the idea of fleet-based exploration is novel

3. The paper has clear results and ablations that justify the validity of its claimed contributions

4. Very promising and timely topic

**Weaknesses**:

1. Missing related work citation on diffusion and online policy attenuation [1]

2. Unclear why the authors focus on continuous control when fleet-based exploration can be a powerful tool in several challenging environments like Craftax [2].

3. Missing comparisons to other exploration algorithms like E3B, Diversity is all you need [3] and planning to explore [4]

4. The evaluation is not complete for acceptance at this point

5. The first few sentences of the abstract are really convoluted for no reason. Strongly suggest rewriting


[1] Frans, Kevin, et al. "Diffusion guidance is a controllable policy improvement operator." arXiv preprint arXiv:2505.23458 (2025).

[2] Matthews, Michael, et al. "Craftax: A lightning-fast benchmark for open-ended reinforcement learning." arXiv preprint arXiv:2402.16801 (2024).

[3] Eysenbach, Benjamin, et al. "Diversity is all you need: Learning skills without a reward function." arXiv preprint arXiv:1802.06070 (2018).

[4] Sekar, Ramanan, et al. "Planning to explore via self-supervised world models." International conference on machine learning. PMLR, 2020.

---

> ### Author Rebuttal · Authors · 2026-03-31
>
> We thank Reviewer Kf7a for their detailed comments and for highlighting the novelty of our work, the clarity of our results, and the promising nature of our approach. Below, we seek to address the reviewer’s listed weaknesses and questions:
>
> **On Additional Related Work:** We appreciate the highlighting of additional relevant works, which we commit to including in the Related Works Section of our updated manuscript. The policy attenuation work (CFGRL) is focused on offline RL, for a policy with *extra conditioning* on optimality or goals, and assumes access to a *given value function* with which to initially label optimality (or some way to provide goals). In contrast, our work focuses on improving exploration during *online* experience collection for sample efficient RL finetuning, for a general BC policy pretrained *without extraneous conditioning terms*, where values are not assumed a priori but *trained* as a critic ensemble on-the-fly from scratch during the learning process.
>
> Diversity Is All You Need is a reward-free exploration strategy that aims to learn multiple diverse policies and skills without environmental supervision. In contrast, the goal of our work is not tabula rasa skill discovery, but the sample-efficient RL finetuning of a pretrained behavior cloning to maximize task success. For such settings we find that DF-ExpEnse achieves superior online exploration for sample-efficient RL finetuning. We will include these related works in our updated manuscript.
>
> **On Continuous Control:** A core contribution of our work is utilizing the multimodal modeling capability of the diffusion model to meaningfully filter the continuous action space to a small but expressive set of reasonable actions, from which it becomes tractable to identify an action with high exploration interest. In contrast, in environments with discrete action spaces such as Craftax, the estimated exploration interest of all possible actions can be directly enumerated at each timestep. Whereas critic ensembles with our proposed fleet-normalization technique can be broadly applicable across environments, we identify diffusion filtering as a crucial component in making such fleet exploration strategies tractable for continuous control tasks.
>
> **On Additional Exploration Algorithms:** We provide additional baseline comparisons across RoboMimic and Gym tasks for E3B and Plan2Explore (P2E); neither of these works have been previously applied to online RL finetuning of pretrained BC models.  As the E3B authors note that the choice of continuous state representation is important, we benchmark against two versions: robot proprioception encoding (E3B-State), and a DINO pixel encoding (E3B-DINO).  Updated plots can be found here ([Comparison Figure](https://sites.google.com/view/dfexpense-rebuttal-figures/#h.8ol5m1oy014c)).  DF-ExpEnse continues to outperform these alternative strategies for evaluating action exploration quality.  Furthermore, DF-ExpEnse does not require additional model components to perform exploration, unlike E3B and P2E.
>
> **On Denoising Steps:** In Appendix A, we provide the denoising steps used in all experiments; 8 for RoboMimic tasks (except Tool Hang which uses 10), and 5 for all Gym experiments.
>
> **On Wall Clock Time and Communication Overhead:** We benchmark wall clock times per step in our [response](https://openreview.net/forum?id=UZF4ssK12K&noteId=DbTJmmnZlk) to Reviewer yojD. Additionally, the intra-fleet communication overhead is virtually negligible, as the only values communicated across the fleet are a small set of floats that summarize ensemble evaluation statistics.
>
> **On Fleet Size:** In Section 4.8 and Figure 5, we provide a study on the effect of using a larger fleet for DF-ExpEnse. We found that performance gains from using a larger fleet size encounters effective saturation past 4 agents. In Appendix B and Figure 6, we further showed that DF-ExpEnse still consistently outperformed baseline approaches across a variety of different fleet sizes; baseline approaches do not utilize fleet scale better than DF-ExpEnse. We provide additional fleet ablations across configurations ([Comparison Figure](https://sites.google.com/view/dfexpense-rebuttal-figures/#h.8s2p3exq0ap)) and against baselines ([Comparison Figure](https://sites.google.com/view/dfexpense-rebuttal-figures/#h.c2rfvpwwh3q8)).
>
> **On Flow Q-Learning:**  FQL is an offline RL technique that trains a flow policy through BC simultaneously with a one-step policy, which is optimized through policy gradient updates and regularized by the flow policy. The core aims are distinct from our work; as the authors of FQL discuss in their limitations section, FQL inherently does not have a “built-in” exploration mechanism for online finetuning. In contrast, DF-ExpEnse explicitly designs a principled *exploration* strategy for collecting high-quality online experience, and seeks to improve the sample efficiency of RL finetuning for offline-pretrained BC policies.

---

> > ### Author Rebuttal · Reviewer_Kf7a · 2026-04-03
> >
> > I will raise my score to a weak accept. The detailed responses were very helpful in clarifying my doubts. However, they also pointed towards the very limited setup the method tackles.

---

> > > ### Author Response · Authors · 2026-04-05
> > >
> > > We thank the reviewer for their increase in score, and are happy to have addressed their questions and concerns.  We would like to add further clarification, which we could not fit in the initial rebuttal response due to the strict character limit and having to respond to multiple questions at once.
> > >
> > > Specifically, in contrast to a limited setup, we hope to highlight the *flexibility* of continuous control as a worthwhile problem to tackle, as well as the *generality* of our proposed approach (DF-ExpEnse) in doing so.
> > >
> > > We focus on continuous control as a challenging, yet practical setting to tackle: practical because robotic decision-making often occurs over continuous action spaces, and challenging because it is intractable to quantifiably evaluate every possible continuous-valued action unlike in discrete settings.  It is also flexible; a variety of task settings are structured as continuous control problems, such as the locomotion (OpenAI Gym), robot arm manipulation (RoboMimic), and dextrous hand manipulation (DexMimicGen) tasks we evaluate on.  We further highlight the general nature of our proposed approach: DF-ExpEnse not only consistently outperforms baselines across this varied set of continuous control tasks and environments, but it also naturally integrates with multiple techniques for the RL finetuning of pretrained BC policies ([DSRL](https://anonymous.4open.science/r/dfexpense_rebuttal_figures-B98C/dfexpense_dsrl_pseudocode.jpg), [Residual RL](https://anonymous.4open.science/r/dfexpense_rebuttal_figures-B98C/dfexpense_resfit_pseudocode.jpg)).  We thus position our work as a highly general exploration technique that can be performantly applied to an expressive class of problems and update strategies.
> > >
> > > We appreciate the reviewer’s thorough feedback in improving our work, and will include this contextualization along with our previous responses and clarifications in our final manuscript.  If the reviewer has any outstanding questions, we would be happy to address them in the remainder of the discussion period.

---

### Official Review · Reviewer_Q4u6 · 2026-03-13

**Soundness:** 3
**Presentation:** 2
**Significance:** 3
**Originality:** 2
**Overall Recommendation:** 4
**Confidence:** 4

**Summary:**

This work considers the problem of exploration in finetuning of BC-trained policies for robotic control. They propose an ensemble-based approach based on the principle of optimism, and couple this with existing RL finetuning approaches to enable more effective exploration. Experimental results on several simulated benchmarks show that this leads to improved performance over finetuning approaches which do not explicitly incentivize exploration.

**Compliance With Llm Reviewing Policy:**

Affirmed.

**Final Justification:**

The authors have added an additional benchmark, and included results running their approach with another RL algorithm (residual RL). This addresses my primary concerns, so I have raised my score accordingly.

**Key Questions For Authors:**

1. In equation (2), why is the first term the min over the Q ensemble rather than, for example, the mean, as is done in the SUNRISE paper [3]?
2. Is a single policy maintained during updates, or an ensemble of policies? Approaches such as SUNRISE maintain an ensemble of policies to increase diversity. This would be an interesting ablation to run.

[3] Lee, Kimin, et al. "Sunrise: A simple unified framework for ensemble learning in deep reinforcement learning." International conference on machine learning. PMLR, 2021.

**Limitations:**

Yes.

**Strengths And Weaknesses:**

**Strengths**
1. Exploration is an important problem in RL, but has largely not been investigated in the context of finetuning robot control policies. This work demonstrates that exploration can lead to improved performance in such settings.

**Weaknesses**
1. The benchmarks the paper considers are relatively limited, and it could be strengthened by including results on additional benchmarks. For example, Libero [1] would be a natural choice. In addition, it would be interesting to consider environments where some amount of exploration is likely needed to find any reward, for example, Antmaze [2].
2. It would also be interesting to run the optimistic approach proposed here with another RL finetuning procedure, such as residual RL.
3. The exposition of the exact algorithm is somewhat unclear. In Section 3.1, it is never stated that the policy itself is updated, though I assume this is the case given the use of DSRL. How is this updated exactly? What is the exact update for the critic? It would be helpful to include an algorithm box explicitly stating the full algorithm.
4. I felt like the emphasis on robot fleets to be somewhat of a distraction from the main contributions of the paper. I would suggest altering the exposition in a way that highlights this less, and frames it more as an implementation detail. This is a matter of personal preference, though, and I do not see this is a critical issue with the paper.

[1] Liu, Bo, et al. "Libero: Benchmarking knowledge transfer for lifelong robot learning." Advances in Neural Information Processing Systems 36 (2023): 44776-44791.

[2] Fu, Justin, et al. "D4rl: Datasets for deep data-driven reinforcement learning." arXiv preprint arXiv:2004.07219 (2020).

---

> ### Author Rebuttal · Authors · 2026-03-31
>
> We thank Reviewer Q4u6 for their detailed comments, and helpful feedback.  We seek to address the listed questions below:
>
> **On Additional Benchmarks and Residual RL Experiments:** We supply additional experimentation on an additional benchmark task suite: DexMimicGen [1].  DexMimicGen features complex bimanual dextrous manipulation tasks, with sparse reward feedback.  Furthermore, to the reviewer’s interest, we provide such experimental results by applying DF-ExpEnse to Residual RL finetuning; namely ResFiT [2], an updated extension of ResiP [3], on pretrained diffusion policies.  We compare the residual-based DF-ExpEnse against vanilla ResFiT and Max-Q baselines, where plots and evaluation videos can be visualized here ([Comparison Figure](https://sites.google.com/view/dfexpense-rebuttal-figures/#h.8b2c1gau74nv)), where all runs are averaged over three random seeds.  We confirm a trend consistent with the DSRL-based implementation findings (Figure 2): that DF-ExpEnse can meaningfully improve sample-efficient self-improvement by enabling more principled exploration compared to alternate approaches.  This highlights the applicability of DF-ExpEnse as a general exploration technique that can be successfully applied across RL finetuning approaches, as well as task settings (locomotion, single-arm continuous control, bimanual dextrous manipulation) and environments.
>
> **On Algorithmic Updates:** DF-ExpEnse is a *general* exploration technique that naturally supports multiple different policy update strategies; as such, the method overview in Section 3.1 only mentions “policy updates” at a high level.  However, for experimental implementation, we do explicitly detail how policy updates are performed when DF-ExpEnse is integrated with DSRL in Section 4.1; in such cases, the RL updates are isolated to the input noise selection policy.  DF-ExpEnse is not limited to only the DSRL update strategy; in the additional Residual RL experiments provided above, the RL updates are isolated to the residual network.  We plan to add additional pseudocode algorithm boxes, previewed here ([Pseudocode Figures](https://sites.google.com/view/dfexpense-rebuttal-figures/#h.u1jlfn8tg5u9)), in our final manuscript.  We also commit to adding textual descriptions on DF-ExpEnse’s policy update strategy when integrated with Residual RL in the final draft, as well as releasing the code upon acceptance.
>
> **On Mean Value Estimation:** We initially opted to use the ensemble minimum, as is commonly done during policy gradient updates during the optimization step, to avoid potential value overestimation.  However, we provide additional experimentation across Can and Square tasks from RoboMimic and Cheetah and Hopper tasks from OpenAI Gym using the ensemble mean for value estimation ([Comparison Figure](https://sites.google.com/view/dfexpense-rebuttal-figures/#h.svxvldbu2tzz)) averaged over three seeds each.  We find that the performance is comparable with using the ensemble minimum.  DF-ExpEnse is therefore robust to this particular design decision.
>
> **On Policy Ensembles:** We clarify that in our experiments, only a single policy is trained and updated; this has the benefit of smaller parameter and computation requirements.  We do agree that exploring an ensemble of policies is an interesting ablation; with the DSRL-based DF-ExpEnse setting, this can be tested by utilizing an ensemble of noise selector policies, as the pretrained diffusion policy is kept frozen.  We run this ablation for a policy ensemble size of 3 and 5 on the Can baseline, and compare it to using a singular noise selection policy with an action candidate size of 3 and 5 (as done in standard DF-ExpEnse), visualized here ([Comparison Figure](https://sites.google.com/view/dfexpense-rebuttal-figures/#h.k7dccaoz4yun)) with each run aggregated over three seeds.  We discover that using policy ensembles achieves competitive performance compared to single-policy DF-ExpEnse; however, standard DF-ExpEnse consistently achieves slightly better performance while also utilizing fewer parameters.
>
> We plan to include SUNRISE [4] as a related work in the final version of the manuscript, as well as provide these empirical ablations on Mean Value Estimation and Policy Ensembles.
>
> [1] Jiang et al., DexMimicGen: Automated Data Generation for Bimanual Dexterous Manipulation via Imitation Learning, ICRA 2025.
>
> [2] Ankile et al., Residual Off-Policy RL for Finetuning Behavior Cloning Policies, arXiv 2025.
>
> [3] Ankile et al., From Imitation to Refinement - Residual RL for Precise Assembly, ICRA 2025.
>
> [4] Lee, Kimin, et al. SUNRISE: A Simple Unified Framework for Ensemble Learning in Deep Reinforcement Learning, ICML 2021.

---

> > ### Author Rebuttal · Reviewer_Q4u6 · 2026-04-01
> >
> > Thank you to the authors for their response. It appears the links included in the rebuttal do not work (they give a 403 error when I try to access them). If the authors could send updated links that do work and include these new results I will consider increasing my score, but without this I will need to keep my score.

---

> > > ### Author Response · Authors · 2026-04-01
> > >
> > > We apologize for the inconvenience; since the rebuttal was released, Google has mistakenly flagged and deleted the temporary email we created to host the Google Site.  We have since hosted the rebuttal figure site at a new link (https://sites.google.com/view/dfexpense-rebuttal/), with a backup anonymous repo here (https://anonymous.4open.science/r/dfexpense_rebuttal_figures-B98C/).
> > >
> > > In particular, the links relevant for Reviewer Q4u6 include:
> > > - **On Additional Benchmarks and Residual RL Experiments:** [Site Figure](https://sites.google.com/view/dfexpense-rebuttal/#h.t8izoi9mc8y5) and [Repo Backup](https://anonymous.4open.science/r/dfexpense_rebuttal_figures-B98C/dexmg_results.png)
> > > - **On Algorithmic Updates:** [Site Figure](https://sites.google.com/view/dfexpense-rebuttal/#h.jgw24tgrimt7) and [Repo Backup DSRL](https://anonymous.4open.science/r/dfexpense_rebuttal_figures-B98C/dfexpense_dsrl_pseudocode.jpg), [Repo Backup ResFiT](https://anonymous.4open.science/r/dfexpense_rebuttal_figures-B98C/dfexpense_resfit_pseudocode.jpg)
> > > - **On Mean Value Estimation:** [Site Figure](https://sites.google.com/view/dfexpense-rebuttal/#h.tzs46bleqj75) and [Repo Backup](https://anonymous.4open.science/r/dfexpense_rebuttal_figures-B98C/ucb_mean_ablation_all.png)
> > > - **On Policy Ensembles:** [Site Figure](https://sites.google.com/view/dfexpense-rebuttal/#h.z1hohvgq0i2x) and [Repo Backup](https://anonymous.4open.science/r/dfexpense_rebuttal_figures-B98C/ensemble_can.png)
> > >
> > > **EDIT:** After a successful appeal with Google customer support, the anonymous gmail account associated with the original rebuttal figure site has been reinstated and all links in the original rebuttal submission should be operational again.
> > >
> > > **EDIT #2**: We thank the reviewer for raising their score and supporting acceptance for our submission.  As authors, we do not have reading permission for any subsequent reviewer comments beyond the Rebuttal Acknowledgement under this ICML discussion format - we hope that we were able to address the reviewer’s questions and comments in their entirety.

---

### Decision · Program_Chairs · 2026-04-30

**Decision:**

Accept (regular)

**Comment:**

The paper applies exploration in the context of online fine tuning robot control policies, and the approach integrates naturally with diffusion policies.  The work demonstrates that exploration can lead to improved performance in this setting. The reviewers and AC agree that this is a useful research direction.

The reviewers appreciate that in rebuttal discussions the authors have added a new benchmark and ran the fine-tuning method with residual RL. The reviewers recommend adding benchmarks and believe this will significantly strengthen the work going forward.

The reviewers note that the proposed method is similar to existing exploration approaches, now applied in the fine tuning regime, and improvements over methods that do not use the exploration are useful but perhaps not overly significant.

Some additional relatively minor concerns revolve around complexity, and general applicability.  The reviewers appreciated the authors rebuttal clarifications and wall clock time comparisons, and encourage the authors to fully develop these going forward.